# Reproducibility Report: Neural Networks Fail to Learn Periodic Functions and How to Fix It

**Mayur Arvind**
BITS Pilani, Goa campus
mayur.arvind387@gmail.com

**Mustansir Mama**
BITS Pilani, Pilani campus
mustansirmama@gmail.com

## Reproducibility Summary

**Scope of Reproducibility**

Neural Networks Fail to Learn Periodic Functions and How to Fix It [15] demonstrates experimentally that standard activations such as $\mathrm{ReLU}$, $\mathrm{tanh}$, $\mathrm{sigmoid}$ and their variants all fail to learn to extrapolate simple periodic functions. The original paper goes on to propose a new activation, which is named the $\mathrm{snake}$ function.

The central claims of the paper are two-fold. (1) The properties of the activation functions are carried over to the neural networks. A $\mathrm{tanh}$ network will be smooth and extrapolates to a constant function, while $\mathrm{ReLU}$ extrapolates in a linear way. Standard neural networks with conventional activation functions are insufficient for extrapolating periodic functions. (2) The proposed activation function manages to learn periodic functions while being able to optimize as well as conventional activation functions. While both experimental proof and theoretical justifications are provided for the claims, we shall only be concerned with testing the claims via experimental means.

**Methodology**

While the author was contacted to clarify certain difficulties, the reproduction of all experiments was completed using only the information provided in the original paper itself. With one exception, the link to all datasets used was also provided in the paper itself. This allowed us to implement most experiments from scratch.

**Results**

We were able to successfully replicate experiments supporting the central claim of the paper, that the proposed $\mathrm{snake}$ non-linearity can learn periodic functions. We also analyze the suitability of the $\mathrm{snake}$ activation for other tasks like generative modeling and sentiment analysis.

**What was easy**

Many experiments included descriptions of the neural network architectures and graphs showcasing performance, giving us a clear benchmark to compare our results against. Links to datasets for all experiments, barring one, were also included in the paper itself.

**What was difficult**

Data for the human body temperature experiment was not available. Proper implementation details were not given for initializing the weights in neural networks with $\mathrm{snake}$ and using $\mathrm{snake}$ with RNNs.

**Communication with original authors**

One author, Liu Ziyin was contacted to provide the dataset used for the human body temperature experiment, elaborate upon the implementation of variance correction during weight initialization and provide his implementation of RNN using $\mathrm{snake}$. He provided the GitHub link to his code for the human body temperature, market index, and extrapolation experiments. He also provided an explanation on how to implement variance correction. While the code for the RNN implementation using the $\mathrm{snake}$ activation was not made public, he provided a screenshot of the same.

# 1 Introduction

Deep neural networks are playing an increasing prominent role in fields as diverse as computer vision [4], speech recognition [2], and language modeling [5]. However, while neural networks are excellent tools for interpolating between existing data, standard versions of these networks are not suited for extrapolation beyond the training range. This causes them to struggle at making predictions in problems with a periodic component.

Previous attempts at addressing neural networks' inability to learn periodic functions have included using periodic activation functions [11, 14]. For example, using $sin(x)$ as the activation function for implicit neural representations has been successful at representing complex natural signals and their derivatives [12]. However in more general cases, experimental results suggest that using $sin$ as the activation function cannot compete against $\mathrm{ReLU}$-based activation functions [10, 6, 1, 13] on standard tasks [7].

The original paper: (1) studies the extrapolation properties of a neural network beyond a bounded region; (2) shows that neural networks with standard activation functions are insufficient to learn periodic functions outside the bounded region where data points are present; (3) proposes a solution for this problem in the form of a novel activation function and its variants, and showcases its performance on toy examples and real-world tasks. We have tested the claims made in the original paper, replicating both the experiments displaying the failure of standard activation functions to learn periodic functions as well as the results of the novel activation function on toy and real-world tasks. We have also conducted experiments of our own to understand how viable the proposed activation function is at replacing existing standards such as $\mathrm{ReLU}$ and $\tanh$.

# 2 Scope of reproducibility

The authors make two key claims:

- Standard neural networks with standard activation functions are insufficient to learn periodic functions outside the bounded region where data points are present.

- The proposed novel activation function can learn periodic functions while maintaining the favorable optimization property of the $\mathrm{ReLU}$-based activations. The novel activation is dubbed "snake":

$$snake_a(x) := x + \frac{1}{a}sin^2(ax)$$

where $a$ is treated as a fixed parameter in initial experiments, and as a learnable parameter in a few experiments. Snake is shown to outperform standard activation functions $\mathrm{ReLU}, \tanh, \mathrm{LeakyReLU}$ [6], as well as more recently proposed functions such as $\mathrm{swish}$ [10], and $\sin$ [12, 7].

Due to the broad and far-reaching consequences of the two claims, the original paper supports them via both theoretical justification and an extensive list of experiments which range from testing performance on toy datasets to real world applications. We have exhaustively replicated the original list of experiments, and have conducted a few additional experiments of our own, using the proposed activation function in a Deep Convolutional Generative Adversarial Network (DCGAN) to generate images of handwritten digits and in a Long Short Term Memory (LSTM) network for sentiment analysis.

# 3 Methodology

The code used by the authors had not been made public at the time we started working on re-implementing the paper. That meant we reproduced all the results in the paper from scratch relying on the descriptions of the neural network architecture and a link or description of the dataset. The descriptions were brief but sufficient such as "feedforward neural network with 2 hidden layers, both with 64 neurons" for the Body Temperature Prediction experiment and "4-layer feedforward network with $1 \rightarrow 64 \rightarrow 64 \rightarrow 1$ hidden neurons" for the Financial Data Prediction experiment. In the case of experiments that utilized large standard networks such as ResNet18, the PyTorch library implementation of the model was used, with $snake$ substituted in place of the default activation functions. Besides the model implementations, we were also required to make $a$ a learnable parameter in $snake$ for a few experiments.

### 3.1 Model descriptions

Models used in the original paper included fully-connected, feed-forward neural networks with different architectures for the various experiments. Larger standard models such as ResNet18 were also used. The authors of the original paper had initially not made their code available and we had to implement most models ourselves.

### 3.2 Datasets

The data used in the extrapolation experiments are directly sampled from periodic functions such as $\sin(x)$. Some experiments dealt with standard datasets such MNIST and CIFAR-10. Data for the real-life datasets had to be downloaded:

- Daily data from 1995-1-1 to 2020-1-31 of Wilshire 5000 Total Market Full Cap Index: Downloaded from link provided in the original paper: `https://www.wilshire.com/indexes`
- Average weekly temperature evolution in Minamitorishima, an island south of Tokyo (longitude: 153.98, latitude: 24.28) after April 2008: Downloaded from link provided in the original paper: `https://join.fz-juelich.de/access`
- Patient body temperature: Made available by the authors upon request
- IMDB Reviews Dataset used for our additional sentiment analysis experiment: Downloaded from `https://www.kaggle.com/lakshmi25npathi/imdb-dataset-of-50k-movie-reviews`

### 3.3 Hyperparameters

Different experiments included varying levels of detail with respect to hyperparameters. Many experiments provided an overview of the neural network architecture (e.g. "4-layer fully connected neural network") but not other hyperparameters, such as batch size, loss function, or learning rate. In cases where information was missing, assumptions had to be made, with some trial-and-error required to obtain a close approximation of the original result. This trial-and-error involved a grid search over the architecture (number of layers, number of neurons in each layer), number of epochs (100 to 5000), batch size (16 to 512), optimizer (Adam, SGD, RMSProp), learning rate (0.001 to 0.1) and value of $a$ in networks with the snake activation (1 to 30).

### 3.4 Experimental setup

The entire codebase has been uploaded to GitHub and is publicly available: `https://github.com/mayurak47/Reproducibility_Challenge`. The experiments were run locally as well as on GPU enabled sessions on Google Colab. All the models and experiments were coded using the PyTorch library.

### 3.5 Computational requirements

Many of the experiments, particularly those relating to regressing different functions and datasets, could be run locally on a MacBook Air with an Intel i5 CPU and 8 GB of RAM, not requiring more than a few minutes to train. The more demanding experiments required the use of GPUs. Training a ResNet18 on CIFAR-10 with six activation functions for 100 epochs took roughly 12 hours on a Tesla T4 GPU on Google Colab. Our additional experiments on training a GAN and an LSTM required roughly 2 hours each on the same hardware.

## 4 Results

Wherever possible, the claims of the original papers were tested and in each case, we were able to reproduce the original results. The list of experiments that we reproduced is listed below.

### 4.1 Extrapolation experiments on analytic functions

Neural networks with a single hidden layer consisting of 512 neurons are trained on data sampled from four different analytic functions using the ReLU and tanh activation functions. The training data is obtained by sampling from [-5, 5] with a gap in [-1, 1]. It is observed in Fig. 1 that the extrapolation of neural networks depends on the activation function used. When ReLU is used, the extrapolation diverges to $\pm\infty$. When tanh is used, the extrapolation levels off. The authors formally prove these observations and conclude that neural networks using these activation functions cannot learn to extrapolate periodic functions.

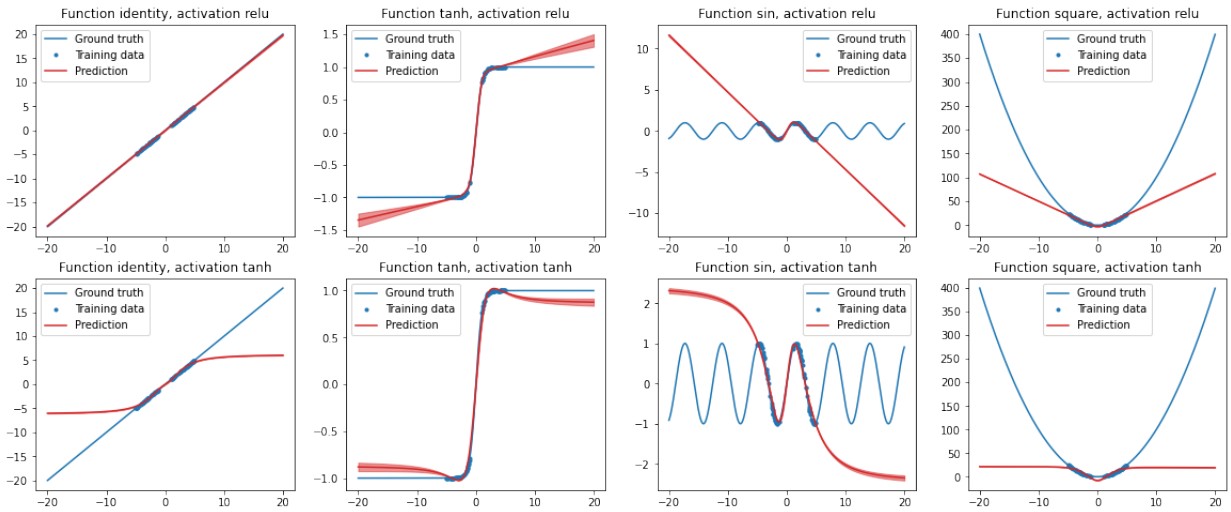

Figure 1: Regressing analytic functions with neural networks having the specified activation function

## 4.2 Applicability of proposed method

It is first demonstrated that the snake activation function is easier to optimize than other commonly used baseline periodic activation functions like $\sin(x)$ and $x + \sin(x)$. Fully-connected neural networks with 3 hidden layers (512 neurons each) are trained on the MNIST dataset. This is a 10-way classification problem, and the training cross-entropy losses for the different networks can be observed in Fig. 2, with the snake network achieving the lowest training loss.

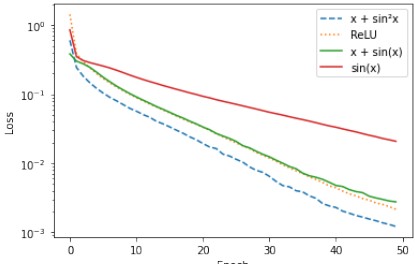

Figure 2: Optimization of different activation functions on MNIST

It is then shown that snake is able to regress the periodic function $\sin(x)$. While all activation functions learn the training data (Fig. 1), only snake is able to capture the periodic behavior of $\sin(x)$ (Fig. 3). The extrapolation diverges from the underlying sin function due to the limited training data used.

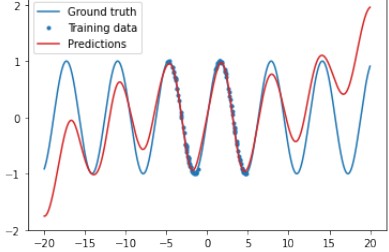

Figure 3: Regressing $\sin(x)$ using the snake activation function

### 4.3 Applications

Multiple experiments are conducted to illustrate the performance of snake on a range of tasks.

ResNet18 [3], with 10M parameters, is trained on the CIFAR-10 dataset. This is a 10-way image classification task. The ReLU layers in the architecture are replaced with the specified activation, and the network is trained for 100 epochs. The LaProp optimizer [1] [16] is used; the learning rate is $4 \times 10^{-4}$ for the first 50 epochs and $4 \times 10^{-5}$ for the next 50. A test accuracy of 93-94% is achieved by the snake network (Fig. 4), in line with that of the other standard activation functions. This suggests that snake is suitable for large-scale image classification problems, and may be used as an straightforward alternative to other activation functions.

The core utility of snake is shown via two real-life problems. The two tasks are predicting the evolution of temperature in Minamitorishima island in Japan (Fig. 5), and the modeling the body temperature of a patient (Fig. 6). The architectures used are $1 \to 100 \to 100 \to 1$ and $1 \to 64 \to 64 \to 1$ respectively, as in the original paper. In the Minamitorishima experiment, the parameters $a$ were made learnable; in the body temperature experiment, $a = 30$. In both cases, snake is the only activation function that makes meaningful extrapolation and predictions. It can also be seen in Fig. 5b that snake is the only activation function that is able to learn the training data - the other non-linearities are unable to fit the training points, irrespective of the number of epochs the models are trained for.

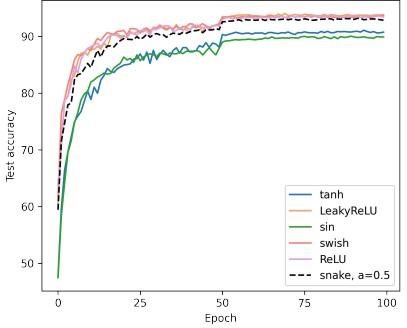

Figure 4: Test accuracy of ResNet18 with different non-linearities

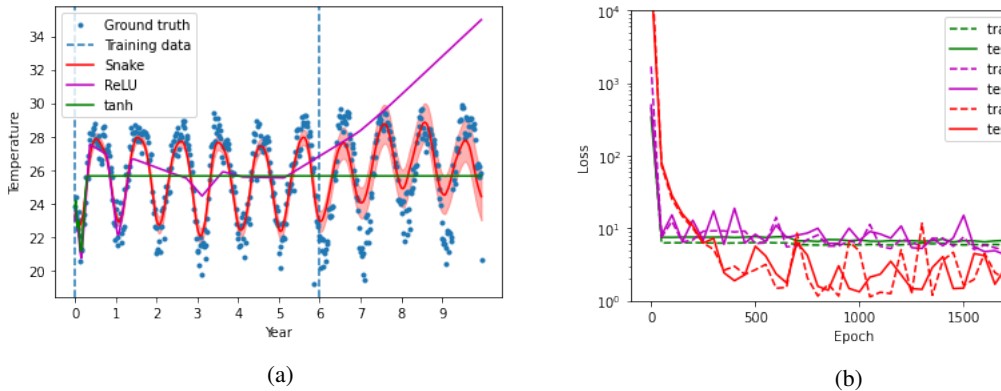

(a)

(b)

Figure 5: Atmospheric temperature evolution. (a) predictions of different networks; (b) train and test losses observed during training

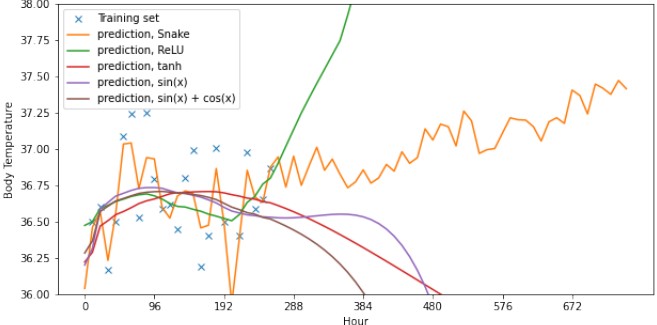

Figure 6: Regressing body temperature

---

[1] Code taken from `https://github.com/Z-T-WANG/LaProp-Optimizer`

The snake network correctly learns the periodicity of the atmospheric temperature dataset, even though the amplitude is slightly off, and correctly infers that body temperature is roughly 37°C.

Another regression problem the authors used to demonstrate the working of snake is that of financial data prediction (Fig. 7). The data used is from the Wilshire 5000 Total Market Full Cap Index, considered representative of the worldwide economic trend. The snake network ( $1 \rightarrow 64 \rightarrow 64 \rightarrow 1, a = 30$ ), which was trained using data from 1995 to 2020-1-31, before COVID-19 impacted the world economy, predicted an economic slowdown in 2020. This might be due to the cyclic nature of world markets, which the model was able to capture. As in the previous regression experiments, snake performs better than conventional non-linearities (Table 1).

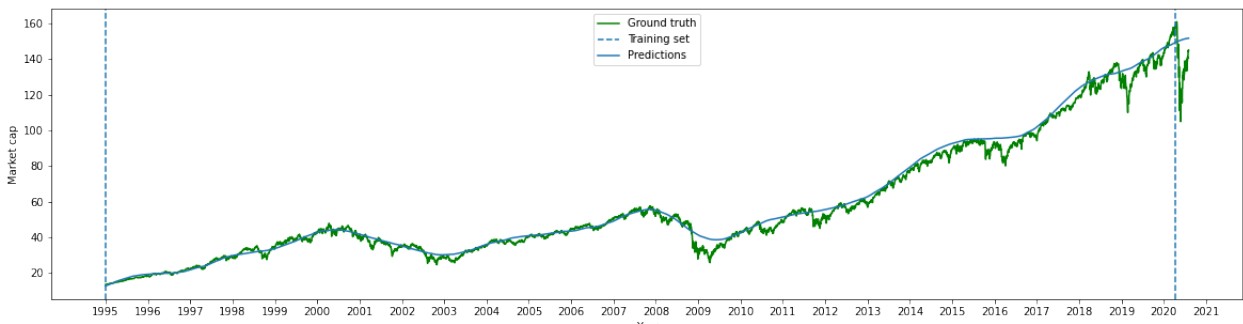

Figure 7: Predicting the Wilshere 5000 index

The authors, in an additional experiment described in the appendix, use this dataset to gain insights into how the snake activation function learns (Fig. 8). Observing the predictions made at various points in the training process, we notice that at first, the features learned are mostly linear, low frequency features are then learned, and high-frequency features are learned at the later stages of training.

| Method | Test MSE |
|---|---|
| Swish DNN | $390.33 \pm 17.57$ |
| ReLU DNN | $343.34 \pm 78.07$ |
| **Snake DNN** | **$211.39 \pm 46.64$** |

Table 1: Prediction of Wilshere 5000 index

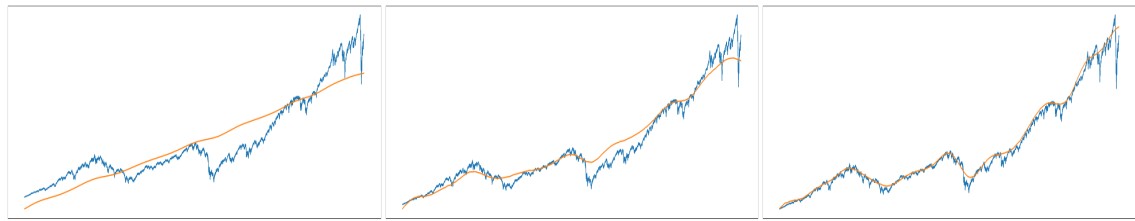

Figure 8: Predictions made by the model after 10, 20, and 50 epochs of training on the Wilshere 5000 index

The performance of a snake feedforward network (two hidden layers of 64 neurons each, $a = 30$) and a recurrent neural network (single recurrent layer, 64 features in hidden state), typically used for time-series prediction, are compared in Fig. 9. The task is to learn the function $\sin(0.1x)$, with Gaussian noise $\sigma$ added, for $T = 300$ timesteps. The first 200 are used for training, while the last 100 are used for testing.

It is seen that because of the noisy training data, even the predictions of the RNN are noisy, with a high generalization loss. The feedforward network, on the other hand, almost perfectly learns the underlying function with the right frequency and amplitude.

Further, RNNs learn by backpropagation through time (BPTT), which has a prohibitively high computation cost, and can result in the exploding/vanishing gradient problem [8]. As a result the time taken by the snake network to regress the function is roughly 2 orders of magnitude lower than the time taken by the RNN (Fig. 10). This suggests that snake networks may be more effective in modeling data that is known beforehand to be periodic in nature.

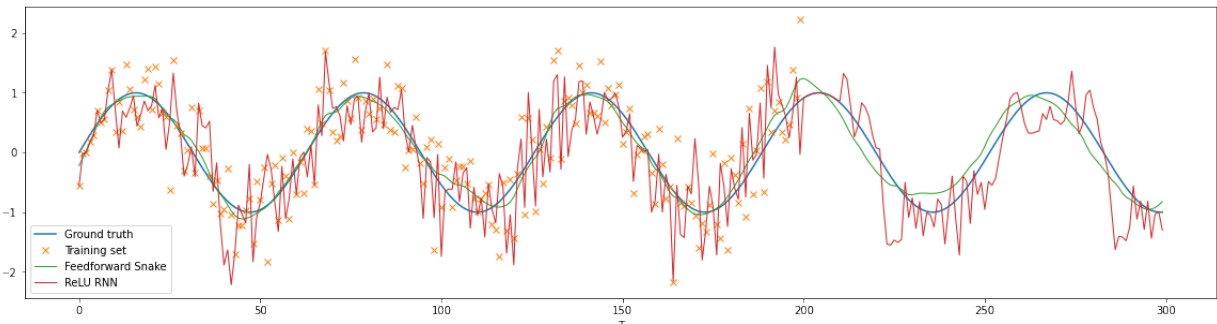

Figure 9: Predictions of a feedforward network with snake activation and a conventional RNN on $\sin(0.1x)$

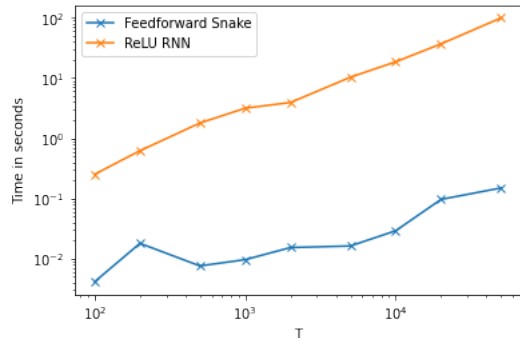

Figure 10: Time taken for a single epoch of training by an RNN and a snake feedforward network on $T$ timesteps of $\sin(0.1x)$

### 4.4 Effect of $a$

In a series of experiments, the authors depict the effect the parameter $a$ has on the learning process. We reproduce one of these experiments for brevity. Simple neural networks ( $1 \rightarrow 64 \rightarrow 64 \rightarrow 1$ ) are trained on the sinusoidal function $\sin(x) + \sin(4x)/4$.

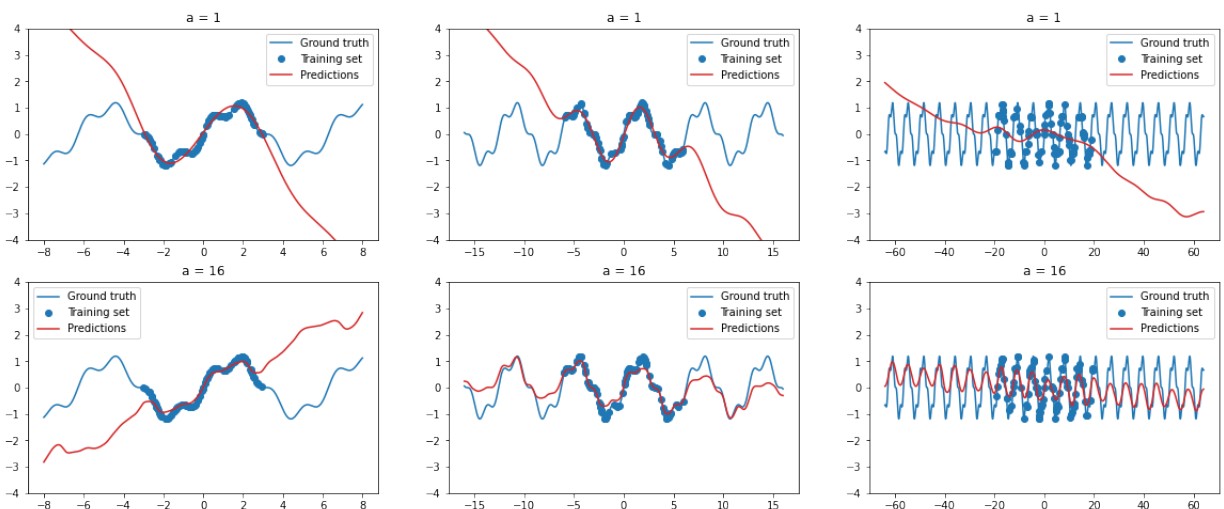

Figure 11: Features learned by snake neural networks at different $a$

182 It is seen in Fig.11 that larger $a$ encourages the model to learn features with higher frequency. With $a = 1$, the higher
183 frequency modulation is considered noise, while the $a = 16$ model learns both the signals. This tendency can be taken
184 into account while working with data known to be periodic, with a well-chosen $a$ speeding up training.

## 4.5 Results beyond original paper

186 The original paper demonstrated the ability of neural networks with the snake activation function to learn periodic
187 functions and that the performance on everyday tasks like image classification is similar to that of conventional activation
188 functions. We extend this study to more sub-fields of deep learning.

189 We train a deep convolutional generative adversarial network (DCGAN) [2] [9] to generate samples of the MNIST dataset.
190 All the activations in the generator and discriminator sub-networks are replaced with the specified non-linearity. We see
191 that while the initial training is slow for the snake GAN (Fig. 13a), it eventually generates realistic samples (Fig. 12a),
192 which are qualitatively indistinguishable from those output by a typical GAN using the LeakyReLU non-linearity (Fig.
193 12b). $a$ was a learnable parameter in this experiment.

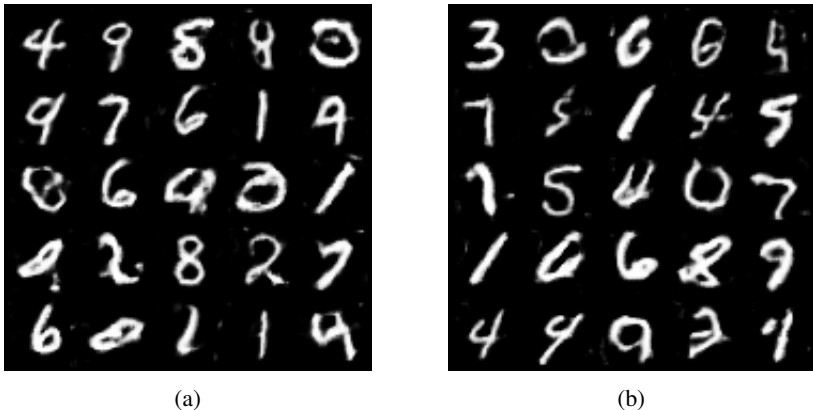

(a)          (b)

Figure 12: Samples output by (a) snake GAN and (b) LeakyReLU GAN after training for 50 epochs

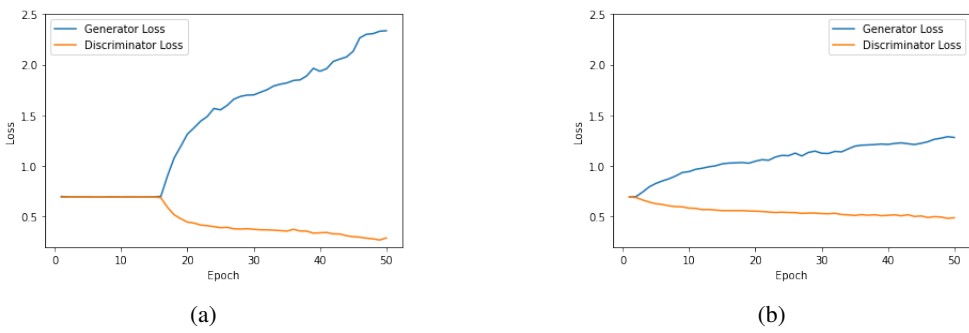

(a)          (b)

Figure 13: Losses observed over the course of training (a) snake GAN and (b) LeakyReLU GAN

194 Finally, we use the snake activation function in a Long Short Term Memory (LSTM) network for sentiment analysis [3]
195 on the IMDB movie reviews dataset. This is a binary classification problem, attempting to predict whether a movie
196 review is positive or negative. The typical tanh activation used to output the value $h_t = o_t * \tanh(C_t)$ in an LSTM
197 is replaced by the snake activation, so that $h_t = o_t * \mathrm{snake}(C_t)$. We observed that the snake LSTM network did not
198 perform very well in this task (Fig. 14) and convergence was much more gradual. A single epoch of training the snake
199 LSTM took twice as long as training the tanh LSTM. Also, in many cases, the snake network got stuck in local minima,
200 necessitating a restart of training.

---

[2]Code adapted from `https://github.com/eriklindernoren/PyTorch-GAN`
[3]Code adapted from `https://www.kaggle.com/arunmohan003/sentiment-analysis-using-lstm-pytorch` and
`https://github.com/piEsposito/pytorch-lstm-by-hand`

A possible explanation for this is that the $\mathrm{snake}$ function is not bounded like $\tanh$, causing an increase in the values of $h_t$. The results of the experiment do not mean that $\mathrm{snake}$ cannot be used in sequence models, only that the application is not as straightforward as in the previous experiments, and further modifications in the architectures might be necessary.

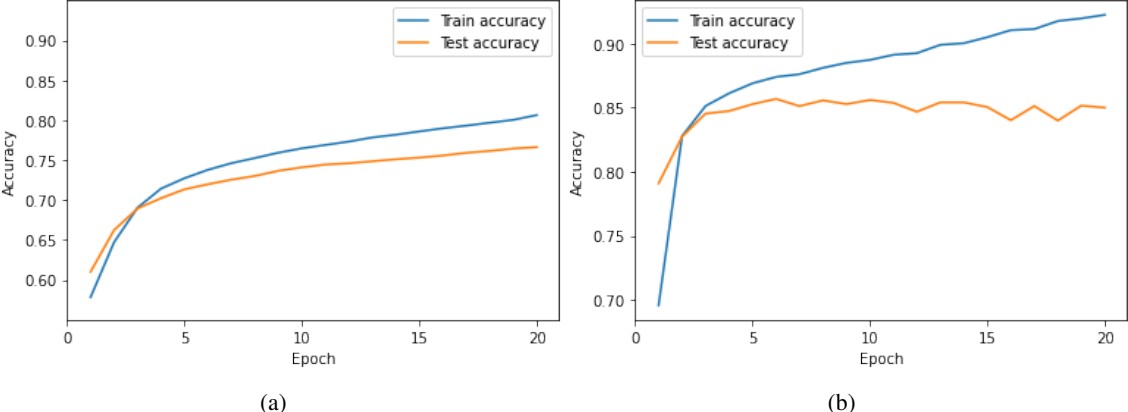

(a)  (b)

Figure 14: Training and testing accuracies versus epochs for (a) $\mathrm{snake}$ LSTM and (b) $\tanh$ LSTM

## 5 Discussion

As the authors had not initially made their code available and only included brief descriptions of the network architectures used in their experiments, exact replication of their experimental results was not possible. However, the qualitative nature of the paper meant that only the relative performance of $\mathrm{snake}$ in comparison to other activation functions on the specified problems was of interest, as opposed to the exact architectural details or loss values achieved. For example, the losses observed in Table 1 and Fig. 5b are orders of magnitude different from those in the original paper, likely due to varying normalization techniques and hyperparameters, even though the overall results observed in Fig. 5a and Fig. 7 are similar to those observed in the original paper. We were able to uphold the claim that neural networks with standard activation functions are insufficient to learn periodic functions outside the training range. We were also able to verify that the proposed activation function performs as well as standard activation functions, $\mathrm{ReLU}, \tanh, \mathrm{LeakyReLU}$, over a wide range of tasks (with the exception of the LSTM experiment), by replicating the experiments in the original paper and conducting some additional ones ourselves. Future work could focus upon providing theoretical justifications for the behavior of $\mathrm{snake}$ and developing more suitable optimization algorithms.

### 5.1 What was easy

A detailed description of the neural network architectures used for experiments such as training on the MNIST dataset and human body temperature was provided, allowing us to replicate the experiments closely. Links to datasets for all experiments, barring one, were also included in the paper itself. An extensive appendix sections listed additional experiments comparing the performance of $\mathrm{snake}$ with different $a$. Every experiment was supported by graphs showcasing the performance of $\mathrm{snake}$ with other activation functions, giving us a clear metric against which we could compare the results of our reproductions.

### 5.2 What was difficult

The original source code was not provided initially and we had to rely on the descriptions of architectures and hyperparameters (which were absent in many cases) and educated guesswork while attempting to replicate the results. Data for the human body temperature experiment was not available. Theoretical justification for variance correction and the results of this variance correction using ResNet101 on CIFAR-10 were provided, but implementation details were not included. The section on Comparison with RNN on Regressing a Simple Periodic Function simply states that $\mathrm{snake}$ was deployed on a feedforward network, without any additional details of the hyperparameters used. The dataset for the experiment had to be inferred from the graphs of the results, and since white noise had been added to the data, exact replication of the experimental setup was not possible.

### 5.3 Communication with original authors

Liu Ziyin, one of the authors, was contacted to provide the dataset used for the human body temperature experiment, elaborate upon the implementation of variance correction and provide the implementation of RNNs using snake.

On being contacted, he provided the GitHub link to his code[4] for the human body temperature, market index, and extrapolation experiments. He also provided an explanation on how to implement variance correction. While the code for the RNN implementation using snake activation was not made public, he provided a screenshot of the same. The provided code was incomplete and not fully documented but was nonetheless valuable in giving us a rough idea about the hyperparameters used. The provided repository also contains the human body temperature dataset within the codebase, which is not available in the original paper.

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
