# OpenReview forum: "Reproducibility Report: Neural Networks Fail to Learn Periodic Functions and How to Fix It"
_ML_Reproducibility_Challenge/2020 — RC2020_

### Official Review · AnonReviewer2 · 2021-02-22
**valuable confirmation**

**Rating:** 7
**Confidence:** 4

**Review:**

The paper is well written and clearly structured.
It is a valuable confirmation of the original paper.
I only have a few formatting criticisms:

sin, cos, and snake should not be italic, but rather \sin or {\rm sin
I think, „4e -4“ should be relaced by 0.0004 or 4\cdot 10^{-4}
In Section 4.4 „Effect of \alpha$, use \boldmath{\alpha}



**Familiar With The Original Paper:**

I have read the original paper

**Reproducibility Summary:**

Report has summary

---

### Official Review · AnonReviewer1 · 2021-02-25

**Rating:** 7
**Confidence:** 4

**Review:**

__Summary__

The original paper (OP) proposes a new activation function, snake. Snake is claimed to be usable with datasets both with and without cyclicity, on the contrary to previous activations. Experiments include low-dimension regression to illustrate the limitations of conventional activations, and applications of snake and baselines to timeseries to illustrate their ability (or not) to deal with cyclic data.
Snake is also evaluated on common benchmarks (CIFAR10) to show that it performs well compared to baselines even when the dataset is not cyclic.
This reproducibility report (RR) reproduces all the experiments of the OP and confirms the OP's findings.
The RR also extends the study to two new settings (image generation and sentiment analysis), both non-cyclic, in which the performance of snake is studied.

__Positive points__
- The OP's code was not publicly available and the authors recoded all of the experiments themselves, which needs to be commended.
- Similarly, the authors optimized hyperparameters to make the implementations work despite the fact that they were missing in the original paper.
- The authors performed additional experiments with elaborate models. Both the feedforward and recurrent architectures that were used were not tested in the original paper, which demonstrates the applicability of the original idea to new settings and  adds value to the original idea from the OP.
- I appreciated the exhaustive replication of the several experiments that this paper contained.
- The RR authors contacted of the OP authors to get additional information on the points of the paper that lacked clarity, and used this information to replicate the OP results.
- The format of the reproducibility report was respected, and the report is overall clear. The authors clearly state that the results are reproducible. The report is overall well written and concise.

__Negative points__
- The general experiments are reproduced, but I find that a finer analysis is absent from the RR. The results are reproduced through a figure, but this figure is often not referred to. A small conclusion about the reproducibility of the specific experiment is systematically absent. No quantified metric on the closeness to the original performance is given, or commented, even though the paper gave quantified results for the financial experiment (OP's Tab.2). For instance, when the performance is illustrated through a graph, I would have expected a comment on the closeness between the accuracy of the OP's model and the RR's model.
- I find that the report lacked precision. For instance: (1) I am not sure which optimizer was used for most of the experiments, or what initialization; Sec 4.1 is an example of this. (2) p.6, l.146: "orders of magnitude": how many? Can you give an estimate of the time in each case? (3) p.7, l.159: can you be more precise than "reasonably realistic samples"? Are there metrics that exist that could quantify by a number the quality of the images produced? (4) p.8, l.163: took much longer: can you be more precise? etc.
- I find that overall, the report does not take a step back w.r.t the experiments it makes. As a consequence, the RR did not reflect on some of the differences, notably Fig.3 (the snake network starts to diverge, on the contrary to OP's Fig. 4) and Fig.5 (the snake network does not fit the training points): how can these differences be explained?
- The RR mentions having to find / guess some hyperparameters (Sec 3.3), but does not comment on the value that was ultimately found and used in the report, or the range of hyperparameters that was tested. It is true that the code is available on Github, but an explicit reference to it would help lift the doubt on certain implementation details. For now, the reader does not gain the knowledge missing from the OP by reading the RR.
- It was often difficult to separate the RR's conclusions from the OP's (for instance, it is confusing on a first read if the last paragraph of p.5 contains conclusions from the RR's authors or the OP's authors, or even if the experiment itself was in the OP (since the experiment does not appear in the main paper, but in the appendix)).

===

Overall, I believe the report could be improved by introducing each experiment better, providing the implementation details, and giving a conclusion on the reproducibility of each experiment. Adding numerical results indicative of reproducibility (was the reproduced end accuracy of the model within X% of the OP's results?) for each experiment would help ground the report, and help it discuss in more details the state of reproducibility of the OP.

===

__Additional comments__
- Some figures are not referenced (ex: Fig.2, Fig.3, Fig.4,). This is problematic because this reinforces the idea that the experiment was run in the RR without an analysis of its results and its significance. How can you tell that the results were indeed replicated?
- The RR mentions that reimplementing the initialization was not obvious from the paper and that one of the OP's authors helped. An additional comment about what was the problem / what was the solution found (or at least a pointer to the RR's code) would be useful.
- Regarding the discussion of Fig.8: it seems to me that the difference between the RNN and the MLP could simply be due to a regularization effect: the RNN also seems to learn the right lower frequency.
- The authors mention in the reproducibility summary that they reimplemented everything from scratch, though looking at the code, it appears that some implementations (of the LaProp optimizer or of the base code for the GAN) were based on existing codebases. While obviously the authors are not expected to recode *everything* from scratch, it would be more transparent to indicate that some parts of the code were built on top of existing code in the RR.
- A mention of the software used (Pytorch for the neural networks) would have been useful, as all libraries do not have necessarily the same default hyperparameters. Similarly, the authors mention in Sec. 3.5 that many experiments could be run locally: more details on the local hardware (which CPU for instance, a minima an indication on the quality of the "local" computer that was used) would be welcome.
- The RR mentions that making the parameter  a learnable was not properly explained in the OP. Here too, a pointer to the solution found would be appropriate.
- The introduction is extremely close to the one in the OP. I would suggest reformulating the introduction more, or possibly shorten it, to avoid repeating the OP.
- The figures were overall well made and close in design to the original paper, which helps when comparing results. Possible improvements could be to respect the color attribution for different models (Fig.2) and making sure that the scale of two plots coincide to make comparisons easier (Fig.10 top left and bottom left, Fig.13 for instance).

**Familiar With The Original Paper:**

I have read the original paper

**Reproducibility Summary:**

Report has summary

---

### Official Review · AnonReviewer3 · 2021-03-10
**Extensive coverage of the experiments corroborating the original work**

**Rating:** 8
**Confidence:** 4

**Review:**

The presented report is well organized, systematic, clear and concise with most of the major experimental results reproduced and reported. To begin with, authors have clearly mentioned the key claims to be investigated in the scope of reproducibility with the focus on the properties of  the activation functions and their behaviour in the experiments reported in the original paper. It has also been proposed to conduct additional experiments beyond what is already covered in the source paper. These includes DCGAN for handwritten digit generation and LSTM model for sentiment analysis, each modified to include the proposed snake activation function. It is however to be noted that when result plots for different cases are not reported in the same graph, it helps to report them on the same scale, wherever possible (e.g. Figure 12 and 13), for easier comparison.

Due to the unavailability of the source code from the author's of original paper, the current authors, in consultation with the original authors, have done a good job of replicating the results, which is matching qualitatively to a great extent with the ones reported in the original paper. While the architectural details were available in the original text, the hyper-parameters and any other information missing were assumed and arrived at with trial and error.

Most of the results from the main content of the original paper has been reported and are found to be qualitatively matching. Figure 1 has demonstrated the inability of learning periodicity outside of the training region and thus substantiates the claim 1 of the scope of reproducibility. These reporting also matches with the corresponding figure in the original work for the widely used activation functions - ReLU and Tanh.

To substantiate claim 2 which states that the proposed snake activation function captures periodicity outside of the training samples and also maintains favourable optimization properties such as loss reduction comparable to or better than ReLU activation, corresponding results from original work has also been reproduced. Although the frequency of the sin wave used in the experiment appears to be different, it is helpful in demonstrating that even though periodicity is observed, significant deviations can still occur at far away places from the training samples. Nevertheless, differences in implementations can be a source of difference.

In addition to the demonstrations on synthetic datasets discussed earlier, the authors have further reproduced results on real-world/application oriented datasets that includes performance reporting of ResNet18 on CIFAR-10 dataset, atmospheric temperature prediction in Minamitorishima island (Japan), patient body temperature prediction and Wilshere 5000 index prediction. All these reported results corroborates the points in the original paper across varying and diverse datasets.

Finally, the authors have reported results for a few experiments from the Appendix section such as the one showing the effect of various value of $a$. It is also admirable that they have reported results and discussed implications of use of snake activation function in other scenarios such as training DCGANs and LSTMs. While the snake activation was shown to have reasonable and comparable training speed and test performances as evidenced by Figure 2 and 4, the extra experiments on non-toy datasets helped in highlighting the relatively slow training speed.

Thus, overall, the authors have tried to cover extensive set of experiments demonstrating pros and cons of the proposed snake activation function and have reproduced the results qualitatively to a great extent. The difficulty in reproducing some of the work such as comparison with RNN on regressing a Simple Periodic function where the details were missing and the data points had to be inferred from the graphs have been mentioned. Similarly, it is also suggested to clarify the implementation details for showing variance correction using ResNet101 on CIFAR-10 for better reproducibility.

**Familiar With The Original Paper:**

I have read the original paper

**Reproducibility Summary:**

Report has summary

---

### Decision · Program_Chairs · 2021-03-31

**Decision:**

Accept

**Comment:**

Selected for ReScience-C Journal Publication.

Good reviews, authors also responded to the concerns and updated their paper accordingly, which makes the report stronger.